# Optimizing the Maize Irrigation Strategy and Yield Prediction under Future Climate Scenarios in the Yellow River Delta

Yuyang Shan [1], Ge Li [1], Shuai Tan [2,*], Lijun Su [1,*], Yan Sun [1], Weiyi Mu [1] and Quanjiu Wang [1]

[1] State Key Laboratory of Eco-Hydraulics in Northwest Arid Region of China, Xi'an University of Technology, Xi'an 710048, China
[2] Faculty of Modern Agricultural Engineering, Kunming University of Science and Technology, Kunming 650500, China
* Correspondence: tan-shuai@hotmail.com (S.T.); sljun11@163.com (L.S.)

**Abstract:** The contradiction between water demand and water supply in the Yellow River Delta restricts the corn yield in the region. It is of great significance to formulate reasonable irrigation strategies to alleviate regional water use and improve corn yield. Based on typical hydrological years (wet year, normal year, and dry year), this study used the coupling model of AquaCrop, the multi-objective genetic algorithm (NSGA-III), and TOPSIS-Entropy established using the Python language to solve the problem, with the objectives of achieving the minimum irrigation water (*IW*), maximum yield (*Y*), maximum irrigation water production rate (*IWP*), and maximum water use efficiency (*WUE*). TOPSIS-Entropy was then used to make decisions on the Pareto fronts, seeking the best irrigation decision under the multiple objectives. The results show the following: (1) The AquaCrop-OSPy model accurately simulated the maize growth process in the experimental area. The $R^2$ values for canopy coverage (*CC*) in 2019, 2020, and 2021 were 0.87, 0.90, and 0.92, respectively, and the $R^2$ values for the aboveground biomass (*BIO*) were 0.97, 0.96, and 0.96. (2) Compared with other irrigation treatments, the rainfall in the test area can meet the water demand of the maize growth period in wet years, and net irrigation can significantly reduce *IW* and increase *Y*, *IWP*, and *WUE* in normal and dry years. (3) Using LARS-WG (a widely employed stochastic weather generator in agricultural climate impact assessment) to generate future climate scenarios externally resulted in a higher $CO_2$ concentration with increased production and slightly reduced *IW* demand. (4) Optimizing irrigation strategies is important for allowing decision makers to promote the sustainable utilization of water resources in the study region and increase maize crop yields.

**Keywords:** AquaCrop-OSPy model; maize; NSGA-III; TOPSIS-Entropy; Yellow River Delta

## 1. Introduction

The Yellow River Delta region has a temperate continental monsoon climate, with significant temperature differences between the four seasons and average annual rainfall of 525–640 mm, where approximately 70% of the annual rainfall is concentrated between July and September [1]. The grain crops in the region are planted following the method of maize–wheat rotation. The planting area of maize was maintained at more than $5.6 \times 105$ ha between 2015 and 2019, with a total output of approximately 4 million tons, of which the planting area of maize in 2019 was $6.6 \times 105$ ha, and the output was 3.97 million tons, accounting for 17.2% of the planting area of maize in 2019 in Shandong Province, and 15.7% of the total output. This region is one of the most important corn production areas in Shandong Province [2]. The maize yield has been severely affected by the growing lack of fresh water resources, uneven spatial and temporal distribution of rainfall, increasing pollution of the water environment, and increased area of saline alkali land [1]. Thus, in order to ensure food security and the sustainable use of land, there is an urgent need to design a reasonable irrigation strategy.

The design of a reasonable irrigation system needs to consider many factors, such as the meteorological conditions, soil conditions, crop characteristics, water quality, and

groundwater level, but it is most important to consider the relationships between crops, water, and nutrients. Numerous studies have aimed to understand the relationships between crops, water, nutrients, and climate, but these types of studies require much time and effort, and thus it would be better if the relationships could be accurately predicted [3]. Many models have been developed to improve the efficiency of agriculture by accurately assessing the relationships between crops, water, nutrients, and atmospheric conditions in order to make predictions. Such crop models used for research include WOFOST [4], DSSAT [5], and CropSyst [6]. These crop models are powerful tools for assessing the minimum *IW*, maximum *Y*, maximum *WUE*, and maximum *IWP* [7,8]. However, these models require many parameters, where some must be acquired from experiments and others are not easy to obtain, thereby causing problems for users [9]. The AquaCrop model was developed by the Food and Agriculture Organization of the United Nations (FAO) in 2009 to address these issues, and it has been widely used because it can effectively describe soil–plant–atmosphere system processes, as well as being simple to operate and user-friendly, with few parameters (only 33 parameters), and thus it has been applied in various practical conditions [10,11]. The model has also been used widely for simulation studies of maize. In particular, Twumasi et al. (2017) simulated the response in terms of the maize yield to climate variations using the AquaCrop model and 22 years of meteorological data by considering four climate change scenarios: mean temperature changes of $\pm 1$ °C to $\pm 3$ °C and mean rainfall changes of $\pm 5$ mm [12]. Donfack et al. (2018) used the AquaCrop model for estimating yields in the northern regions of Cameroon in the dry and rainy seasons. They estimated the yields in these seasons based on the irrigation volume, where they showed that irrigating maize with 72.15 mm of water in the rainy season could increase the yield by 0.24 t/ha and irrigating with 427.03 mm in the dry season could increase the yield by 1.22 t/ha [13]. Martini (2018) simulated rainfed maize yields using climate data from 1987 to 2016 in southern Brazil and analyzed the sensitivity of parameters such as the soil water stress level, maximum effective rooting depth, root zone crop coefficient, groundwater recharge, and planting density, which are required by the AquaCrop model. The results show that the crop cycle duration, planting density, and field practices had minimal effects, whereas the root zone crop coefficient, *WUE*, soil water storage, and groundwater recharge were the parameters with the greatest effects on rainfed maize yields [14]. In addition, in order to improve the effectiveness of the model, ACOSP is an open-source Python implementation of AquaCrop developed by Kelly and Foster, which can be implemented for integration with other Python modules [15].

Irrigation systems are mainly optimized using the dynamic programming method in the early stage, but the complexity of the solution increases rapidly with the number of state variables and the number of stage divisions [16]. Thus, irrigation system optimization gradually shifts from a single-objective problem to a multi-objective problem, and the optimization results obtained by the multi-objective optimization algorithm generate a set of non-dominated options by analyzing the trade-offs between objectives based on the Pareto front [17]. Huo and Hang (2007) developed an irrigation system optimization model by using the irrigation date as the decision variable and maximum relative yield as the decision objective [18]. Qie et al. (2011) used the irrigation date and irrigation water amount as decision variables, and set the maximum relative crop yield and total irrigation water amount for the whole crop reproductive period as the optimization objectives [19]. At present, the non-dominated sorting genetic algorithm III (NSGA-III) based on the original NSGA-II for solving the multi-objective optimization problem is mainly used for the multi-objective optimization of irrigation strategies. Compared with NSGA-II, NSGA-III overcomes the problem of simultaneously optimizing more than two objectives in NSGA-II and the defects due to the poor diversification of the Pareto front [17].

The objectives of this study were (1) to construct four optimal irrigation schemes based on the AquaCrop-OSPy crop model and NSGA-III framework to minimize *IW* and maximize *Y*, *WUE*, and *IWP* for maize in the Yellow River Delta region in wet years (defined as the 75% frequency precision), normal years (50% frequency precision), and dry

years (25% frequency precision) [20]; (2) to solve the multi-objective optimization problem for maize in the Yellow River Delta region of China; and (3) to determine the optimal irrigation strategy.

## 2. Materials and Methods

### 2.1. Study Region

The trial area for the maize (Jinan 30 variety) field trials from April 2019 to October 2021 was located in the high-tech demonstration base of Dongying Agricultural High-tech Zone, Shandong Province, China (37°21′ N, 118°57′ E, elevation 14 m) (Figure 1). This region has a temperate continental monsoon climate with an average annual rainfall of 587.4 mm, which is mostly concentrated in June to September, an average temperature of 12.3 °C, an annual average of 2234 sunshine hours, and an average frost-free period of 198 days. Maize was sown on 25 April in 2019 and harvested on 6 September. In 2020 and 2021, maize was sown on 15 June and harvested on 7 October. Solar radiation, temperature, humidity, and wind speed meteorological data were monitored during the field trials in 2019–2021 using an automatic weather station (Weather Hawk 500, Campbell Scientific, Logan Utah, UT, USA), which was installed at the base (Figure 2). Minimum and maximum temperatures, rainfall, and reference evapotranspiration for the period of 1961–2015 were obtained from the China Meteorological Data Service Center (http://data.cma.cn (accessed on 26 March 2016)). Three irrigation treatments (I1, I2, and I3) were tested according to the maize growth status, with total irrigation of 186.3 mm under I1 total irrigation of 130 mm under I2, and rainfed conditions under I3, with three replicates for each treatment. The maize growth period was divided into the seedling stage, jointing stage, tasseling stage, and filling stage, and the irrigation measures for the treatments in 2019–2021 are shown in Table 1. The properties of the soil in the 0–80 cm soil layer in the test area are shown in Table 2.

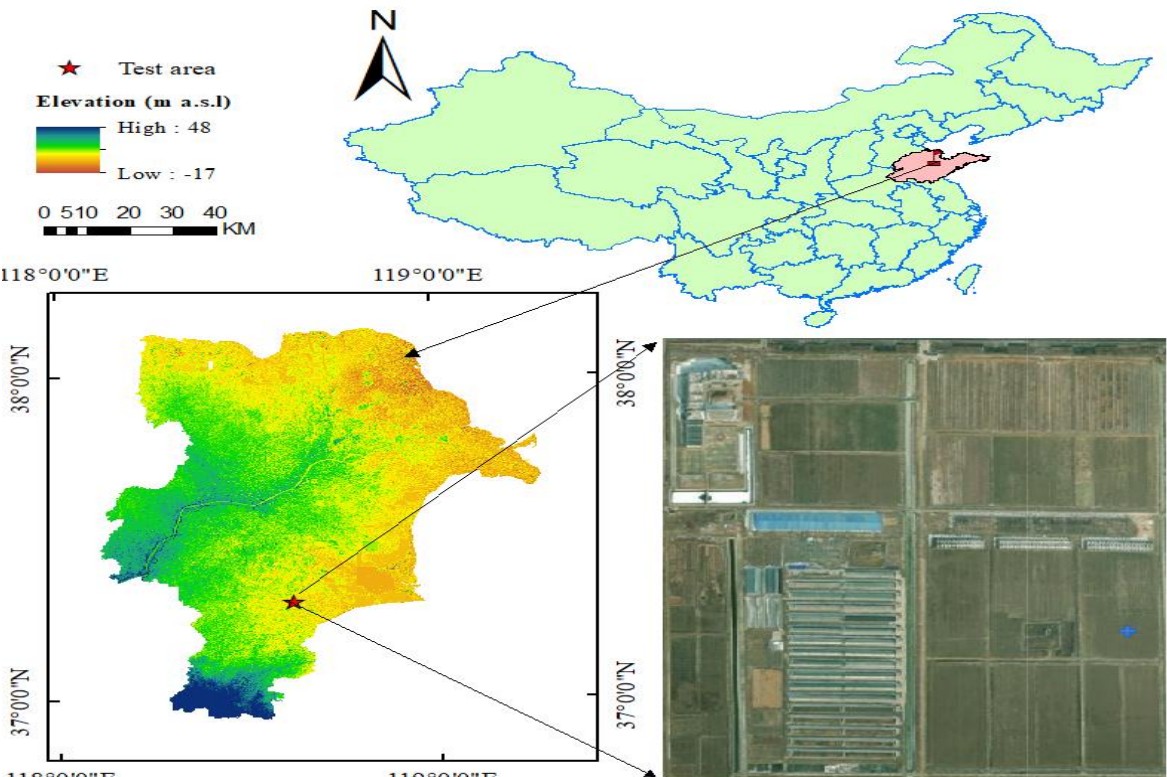

**Figure 1.** Yellow River Delta modern agriculture experimental demonstration base of Shandong Academy of Agricultural Sciences, Dongying City, Shandong Province, China.

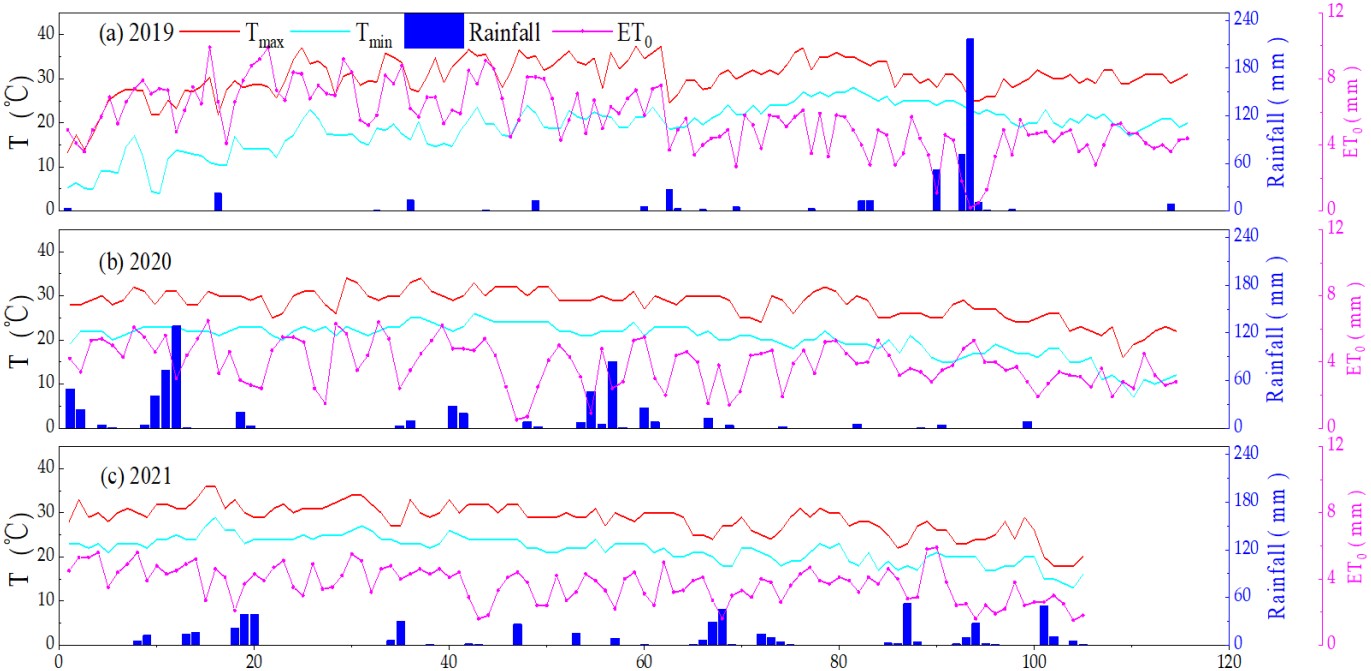

**Figure 2.** Meteorological data during the maize growth periods in 2019–2021 (**a**–**c**). Tmax and Tmin are the maximum and minimum air temperature.

**Table 1.** Irrigation treatments in the maize-growing seasons during 2019–2021.

| Year | Sowing Date | Irrigation Amount (mm) | | | |
|------|-------------|----------|----------|---------|-------|
| | | **Seedling** | **Jointing** | **Filling** | **Total** |
| 2019 | 25 April | 21.6 | 74.7 | 90.0 | 186.3 |
| 2020 | 25 June | 15.2 | 52.3 | 62.5 | 130.0 |
| 2021 | 25 June | 0 | 0 | 0 | 0 |

**Table 2.** Soil properties at the field site.

| Soil Layer (cm) | Soil Texture | | | | Bulk Density (g/cm³) | SAT (cm³/cm³) | FC (cm³/cm³) | PWP (cm³/cm³) | Ks (mm/d) |
|-----------------|-----------|----------|----------|--|----------------------|---------------|--------------|---------------|-----------|
| | **Clay (%)** | **Silt (%)** | **Sand (%)** | | | | | | |
| 0–10 | 3.4 | 24.5 | 72.1 | Sandy loam | 1.40 | 0.490 | 0.247 | 0.050 | 59.8 |
| 10–20 | 2.2 | 18.4 | 79.4 | Sandy loam | 1.46 | 0.490 | 0.247 | 0.050 | 59.8 |
| 20–40 | 2.9 | 21.8 | 75.3 | Sandy loam | 1.51 | 0.530 | 0.218 | 0.043 | 62.7 |
| 40–60 | 6.7 | 55.4 | 37.9 | Silty loam | 1.54 | 0.530 | 0.218 | 0.043 | 62.7 |
| 60–80 | 7.6 | 50.9 | 41.5 | Silty loam | 1.56 | 0.530 | 0.300 | 0.045 | 46.9 |

SAT: soil water content at saturation; FC: field capacity; PWP: permanent wilting point; and $K_s$: saturated hydraulic conductivity.

### 2.2. Determination of Typical Hydrologic Scenarios

The probability distribution of Pearson III, representing a generalized gamma distribution, is suitable for presenting the hydrological pattern of China and was used for determining the different hydrologic scenarios [11,20]. The probability density function is as follows:

$$f(x) = \frac{\beta^{\alpha}}{\Gamma(\alpha)}(x - \alpha_0)^{\alpha-1}e^{-\beta(x-\alpha_0)} \tag{1}$$

where $\alpha$, $\beta$, and $\alpha_0$ are the shape, scale, and location parameters, respectively.

In this study, three hydrological scenarios were developed considering the Pearson III probability distribution: dry year, normal year, and wet year. The Pearson type III distribution method is expressed as follows:

(1)  First, the historical precipitation maize growing seasons (Figure 3) are arranged in descending order ($X_1$, $X_2$, $\cdots$, $X_m$, $\cdots$, $X_n$), where $n$ is the number of the growing seasons, 55.

(2)  Each descending order is assigned a value of $m$, where $X_m$ represents that the number of descending orders greater than or equal to $X_m$ is $m$.

(3)  The accumulated frequency of each growing season ($P$) of the Pearson type III distribution is calculated as follows [21]:

$$P = \frac{m}{n+1} \times 100\% \tag{2}$$

(4)  According to the precipitation frequency, $n$ observations were divided into dry year, normal year, and wet year, and the corresponding precipitation frequency was 25%, 50%, and 75%, respectively.

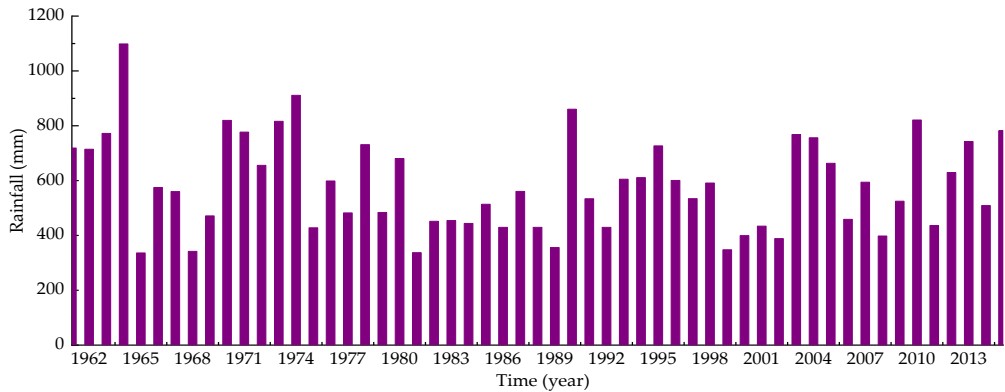

**Figure 3.** Average rainfall in historical years from 1961 to 2015.

The growth seasons 1961, 1993, and 1982 are taken as the wet scenario, normal scenario, and dry scenario, respectively. The precipitation amounts during the growing season are 546.7 mm, 380.7 mm, and 291.0 mm, respectively. Figure 4 shows the specific climate conditions in these three growing seasons.

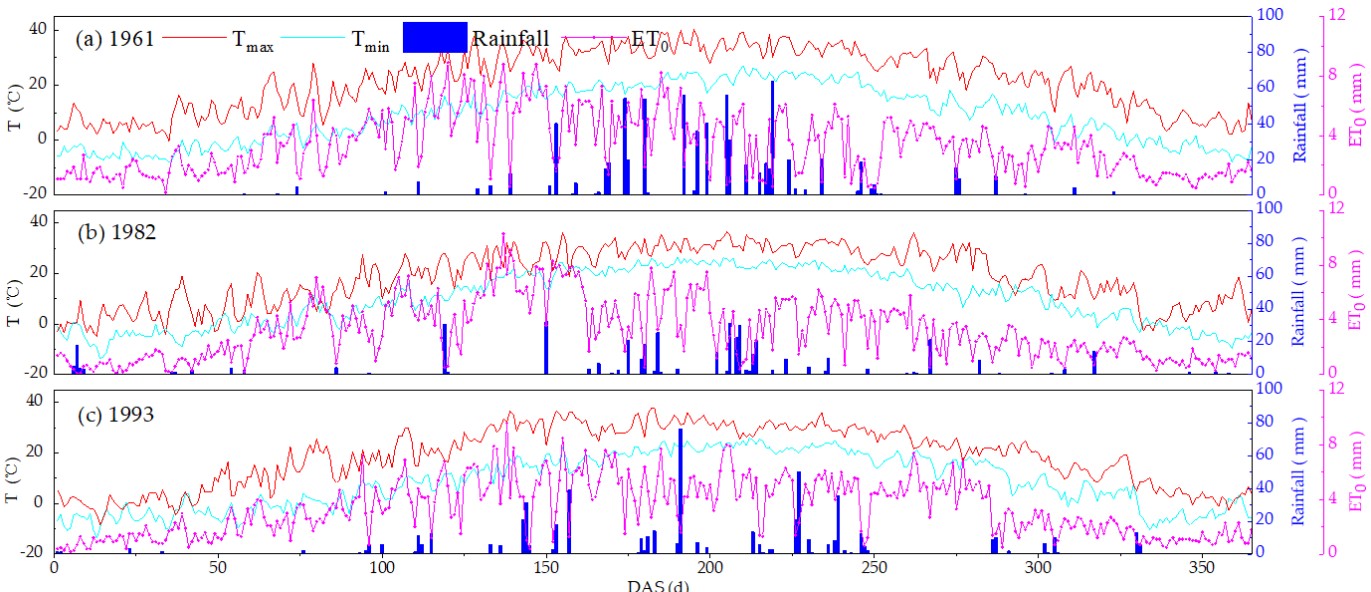

**Figure 4.** Specific climatic conditions in typical hydrological years. Tmax and Tmin are the maximum and minimum air temperature.

### 2.3. Description of the AquaCrop Model

The AquaCrop model is used as a crop growth model to simulate the relationships between the growth process, yield, and water for different crops, with fewer input parameters and a more readily accessible interface compared with other growth models [20,22]. Studies have shown that the AquaCrop model can accurately simulate crop yields under various irrigation scenarios, and the model has been used to successfully simulate the growth process, yield, and water use for maize, wheat, rice, and cotton crops [23,24]. The AquaCrop simulation focuses on four processes: *CC*, crop transpiration (*Tr*), *BIO*, and *Y*. *CC* is converted from leaf area index (*LAI*) using Equation (3), and the leaf area index of maize is obtained by measuring the length of the green leaf leading to the top and the maximum leaf width of maize and multiplying them by a correction factor of 0.75 [25], with the following core formulae [23].

The *CC* is calculated based on the leaf area index (*LAI*) with the following formula:

$$CC = 1 - \exp^{(-0.65 \times LAI)} \tag{3}$$

The $T_r$ is calculated as

$$T_r = K_{S_{Tr}} \times K_{Sw} \times CC^* \times K_{C_{Tr,x}} \times ET_0 \tag{4}$$

where $K_{STr}$ is the temperature stress coefficient, $K_{Sw}$ is the water stress coefficient, $CC^*$ is the adjusted green canopy cover, $K_{CTr,x}$ is the maximum crop transpiration coefficient, and $ET_0$ is the reference crop evapotranspiration.

The *BIO* is calculated as

$$BIO = WP^* \times \sum \frac{T_r}{ET_0} \tag{5}$$

where *WP** is the standardized moisture productivity.

The *Y* is calculated as

$$Y = f_{HI} \times HI_0 \times B \tag{6}$$

where $f_{HI}$ is the adjustment factor and $HI_0$ is the harvest index.

### 2.4. Evaluation Indicators for Effectiveness of Simulations

The effectiveness of using the AquaCrop-OSPy model for simulations was evaluated with statistical indicators comprising the normalized root mean square error (*NRMSE*), coefficient of determination ($R^2$), and Nash–Sutcliffe efficiency coefficient (*NSE*). *NRMSE* values of 10–20% are considered to indicate the good performance of model simulations and *NRMSE* values of 20–30% denote fair model performance [26–29]. The *NSE* values range from $-\infty$ to 1, and the ideal value is *NSE* = 1. The values are calculated as

$$NRMSE = \sqrt{\frac{\sum_{i=1}^{n} (S_i - O_i)^2 / n}{\overline{O}}} \times 100\% \tag{7}$$

$$R^2 = \left( \frac{\sum_{i=1}^{n} (O_i - \overline{O})(S_i - \overline{S})}{\sqrt{\sum_{i=1}^{n} (O_i - \overline{O})^2 \sum_{i=1}^{n} (S_i - \overline{S})^2}} \right)^2 \tag{8}$$

$$NSE = 1 - \frac{\sum_{i=1}^{N} (O_i - S_i)^2}{\sum_{i=1}^{N} (O_i - \overline{O})^2} \tag{9}$$

where $O_i$ is the observed value and $S_i$ is the simulated value, $\overline{O}$ is the average of the measured values, $\overline{S}$ is the average of the simulated values.

*2.5. Scenario Simulation*

2.5.1. Irrigation Scenario Settings

In this paper, six different irrigation strategy scenarios of T1–T6 were set up to simulate the maize *IW*, *Y*, *IWP*, and *WUE* under different scenarios [22]. T1, T3 and T4 are set according to the irrigation plan tested in 2019–2021. T1 is rain-fed irrigation, and the total irrigation volume of T3 and T4 is 180 mm and 130 mm, respectively. The T2 irrigation scenario setting is to define the threshold value ($TAW_j$ (%)) of the main growth period of maize, so that once the soil water content in the growth process of corn is lower than the threshold value, irrigation will be triggered. T5 is net irrigation. According to the water consumption threshold of the maize root zone, the total water content of the crop root is kept above the total available water (*TAW* (%)) through irrigation measures. According to the research of Kelly et al. (2021), this specific threshold is set at 70% [15]. The T6 scenario setting is based on the analysis of meteorological data, and the soil water content *TAW* (%) is set. When the soil water exceeds *TAW* (%) or there is no rainfall in the next ten days, irrigation will be triggered. These six irrigation scenarios are summarized in Table 3.

**Table 3.** Settings for six irrigation scenarios in ACOSP.

| Scenario | Settings |
|---|---|
| T1 | Rain-fed condition |
| T2 | Optimal $TAW_j$ (%) irrigation [1] |
| T3 | The total amounts are 180.0 mm |
| T4 | The total amounts are 130.0 mm |
| T5 | Net irrigation |
| T6 | Optimal irrigation under weather conditions [2] |

[1] *j* represents the thresholds of four major maize growth stages (emergence, canopy growth, max canopy, and senescence). [2] If no rain occurs for the next 10 days, or if rain occurs in the next 10 days but the soil is over 70% depleted, 10 mm irrigation is applied; otherwise, no irrigation is applied.

Scenario T2 defines four thresholds for the total available water ($TAW_j$, %) (for *j* = 1, 2, 3, 4) for crops in the seedling emergence, canopy growth, maximum canopy, and senescence periods. These four thresholds represent *x* [0], *x* [1], *x* [2], and *x* [3] in the decision variables, and the decision variable *x* [4] is *IW* (mm). In the crop growing season, if the soil water content is lower than the specified threshold, irrigation is triggered until the soil water content reaches the field capacity. In the multi-objective optimization problem for scenario T2, the minimum *IW*, maximum *Y*, maximum *IWP*, and maximum *WUE* are used as the objective functions. The objective function expression, decision variables, and related constraints are defined as follows [22]:

$$Objectives \begin{cases} \min IW = \sum_{i=1}^{n} I_i \\ \max Y = f(TAW_j(\%), IW) \\ \max IWP = \frac{(CY_{irr} - CY_{rain-fed}) \times 100}{IW} \\ \max WUE = \frac{CY_{irr}}{ET \times 10} \end{cases} \qquad (10)$$

$$Constraints \begin{cases} I_{\max} > I_i > I_{\min}(i \le n, i \in N^*) \\ TAW_{\max} > TAW_j > TAW_{\min} \end{cases} \qquad (11)$$

where *IW* is the total irrigation volume (mm) for the whole irrigation season and *i* is the number of times that irrigation is applied; $TAW_j$ (%) is the threshold value of the total effective water for each major growth period and *j* is used to differentiate between fertility periods; $CY_{irr}$ is the crop yield under different irrigation conditions (kg/ha), and $CY_{rain-fed}$ is the crop yield under rainfed conditions (kg/ha); *ET* is the total daily evapotranspiration for winter maize during the growing season (mm); $I_{\max}$ is the upper limit of each irrigation amount, $I_{\min}$ is the lower limit of each irrigation amount, and *i* is the number of times irrigation is applied; *j* is used to differentiate between the maize growth stages; $TAW_j$ is the

effective water threshold (%) for each remaining growth stage; and $TAW_{\text{max}}$ and $TAW_{\text{min}}$ comprise the depleted soil water content levels in the allowed range (%).

Scenario T6 is a user-defined irrigation strategy based on meteorological data, where the functions for irrigation are defined according to the following logic:

a. If no rain occurs in the next 10 days, the irrigation amount is $x$ [0].
b. If rain occurs in the next 10 days but the soil moisture is depleted by more than $x$ [2], the irrigation amount = $x$ [1].
c. Otherwise, irrigation amount = 0.

The four optimization objectives in the multi-objective optimization problem established for T6 are the same as those for T2, where $x$ [0], $x$ [1], and $x$ [2] are decision variables. In Equation (15), $I_{no\text{-}rain}$, $I_{rain}$, and $TAW$ (%) correspond to $x$ [0], $x$ [1], and $x$ [2], respectively [22].

$$
objectives\begin{cases} \min IW = \sum_{i=1}^{n} I_i \\ \max Y = f(TAW(\%), I_{rain}, I_{no-rain}) \\ \max IWP \\ \max WUE \end{cases}
$$

$$
Constraints\begin{cases} I_{\max} > I_i > I_{\min}(i \le n, i \in N^*) \\ TAW_{\max} > TAW_j > TAW_{\min} \end{cases}
$$

(12)

where $I_{rain}$ (mm) is the irrigation amount under the condition that rain occurs in the next 10 days but $TAW$ (%) is depleted by more than $x$ [2] and $I_{no\text{-}rain}$ (mm) is the irrigation amount under the condition that no rain occurs in the next 10 days.

### 2.5.2. Solution Based on the NSGA-III Algorithm

The multi-objective optimization model used in this study has four objective functions and three constraint conditions. In order to obtain a reasonable global optimal solution, we used the NSGA-III algorithm with good performance in the multi-objective optimization problem in this study. The NSGA-III algorithm is based on NSGA-III, proposed by Jain and Deb (2014), where a reference point mechanism is introduced. The algorithm deals with multiple optimization objectives by reserving the non-dominated population of individuals close to the reference point [17].

In order to deal with the constraints on Pareto dominance, we used the constraint violation (*CV*) value to quantitatively describe the degree of violation. If a certain solution x satisfies the constraint conditions, it is called a feasible solution, but an infeasible solution if it does not. The *CV* value for the infeasible solution is calculated as [30]

$$
CV(x) = \sum_{g=1}^{G} |g_g(x)| + \sum_{k=1}^{K} |h_k(x)|
$$

(13)

where $G$ is the number of inequality constraints, $K$ is the number of equality constraints, $G$ (x) is an inequality constraint, $H$ (x) is an equality constraint, and $G$ (x) is an inequality constraint. When $G$ (x) < 0, $G$ (x) = −$g$ (x); otherwise, $g$ (x) = 0. When $x$ satisfies any of the constraint conditions, i.e., $x$ is within the feasible region, $CV$ (x) = 0. When $x$ does not completely satisfy the constraint conditions, i.e., $x$ is not in the feasible region, then $CV$ (x) ≠ 0. When the *CV* value is smaller, $x$ is closer to the feasible region.

The hypervolume (*HV*) is used as an index in order to evaluate the performance of the NSGA-III algorithm. *HV* represents the volume of the region in the target space enclosed by the non-dominated solution set and the reference point obtained by the algorithm. When the *HV* value is larger, the comprehensive performance of the algorithm is better and the distribution of the solutions in the optimal solution set is more uniform in the target space. *HV* is calculated as [31].

$$
HV = \sigma \times U_{c=1}^{|s|}(v_c)
$$

(14)

where $\sigma$ represents the Lebesgue measure used to measure the volume, $|s|$ is the number of non-dominated solution sets, and $v_c$ is the super-volume formed by the reference point and the $c$-th solution in the Pareto solution set.

The spacing metric is a measure of the standard deviation of the minimum distance from each point on the Pareto front relative to other points obtained by the multi-objective optimization algorithm, and it is used to measure the uniformity of the solution set. The spacing metric is calculated as follows:

$$Spacing = \sqrt{\frac{1}{X_1 - 1} \sum_{x=1}^{X_1} \left(\bar{d} - d_x\right)^2} \qquad (15)$$

where $X_1$ is the number of midpoints of $P$, $d_x$ represents the minimum distance from point $x$ in $P$ to other points, and $\bar{d}$ is the average of all $d_x$. The spacing metric is inversely proportional to the uniformity of the solution obtained by the algorithm. If the spacing value is 0, all points on the Pareto front obtained by the algorithm are equidistant.

Based on the NSGA-III optimization algorithm, a multi-objective optimization model for maize crop growth in the Yellow River Delta region of Shandong Province, China was constructed. The irrigation volume and *TAW* (%) were selected as the optimization decision variables for NSGA-III coupled with the AquaCrop-OSPy model, and *IW*, *Y*, *IWP*, and *WUE* were the optimization objectives. After a large number of experiments, the population number was set to 500 for T2 and T6, and the number of iterations to 200. The specific parameters used in the experiment are shown in Table 4, which are similar to those used in [32]. The simulations were conducted using a PC with an Intel (R) Core (TM) i5-10300H CPU @ 2.50 GHz and 16 GB RAM.

**Table 4.** Parameters used for optimization with NSGA-III.

| Parameter | Scenario T2 | | | Scenario T6 | | |
|---|---|---|---|---|---|---|
| | Wet Year | Normal Year | Dry Year | Wet Year | Normal Year | Dry Year |
| Number of decision variables | 5 | 5 | 5 | 3 | 3 | 3 |
| Number of objective functions | 4 | 4 | 4 | 4 | 4 | 4 |
| Population size | 500 | 500 | 500 | 500 | 500 | 500 |
| Crossover probability | 0.2 | 0.2 | 0.2 | 0.2 | 0.2 | 0.2 |
| Mutation probability | 0.01 | 0.01 | 0.01 | 0.01 | 0.01 | 0.01 |
| Number of iterations | 200 | 200 | 200 | 200 | 200 | 200 |
| Evaluation number | 91,000 | 91,000 | 91,000 | 91,000 | 91,000 | 91,000 |
| Execution time (s) | 6720.44 | 6821.24 | 5948.56 | 14,038.52 | 14,259.07 | 14,222.88 |
| Number of non-dominated solutions | 455 | 455 | 455 | 455 | 455 | 455 |
| *HV* | 0.005 | 0.009 | 0.010 | 0.013 | 0.040 | 0.040 |
| Spacing | 0.814 | 0.360 | 0.577 | 1.092 | 6.110 | 2.814 |

### 2.5.3. Scheme Optimization of TOPSIS-Entropy Comprehensive Evaluation Model

The TOPSIS is a common evaluation method based on multiple objectives that can fully utilize the original data and accurately reflect the differences between evaluation schemes. TOPSIS is one of the most widely used multi-criteria decision-making methods [31]. According to the principle of the entropy weight method, when the degree of variation in the index is smaller, the amount of information represented is smaller and the corresponding weight is lower. The method for calculating the evaluation object and positive and negative ideal solutions is improved in the TOPSIS-Entropy comprehensive evaluation model in order to make the evaluation results consistent with real situations. The computational steps in the TOPSIS-Entropy model are as follows [33].

(1)  Let $W = (w_1, w_2, w_3, w_4)$ be the relative weight vector of each target calculated by the entropy weight method, which satisfies

$$\sum_{j=1}^{n} \omega_j = 1 (j = 1, 2, 3, 4) \tag{16}$$

where *j* is the number of objectives for the multi-objective optimization problems.

(2) $X_{ij}$ is a solution on the Pareto front and $X_{ij}$ represents the *i*th solution on the *j*th objective function. To normalize the objective function values, the following equation is used:

$$r_{ij} = \frac{x_{ij}}{\sqrt{\sum_{i=1}^{m} x_{ij}^2}} (i = 1, 2, \cdots, n, j = 1, 2, 3, 4) \tag{17}$$

(3) A weighted objective function normalization matrix is calculated according to the weights obtained in the first step as

$$V_{ij} = w_j \times r_{ij} (i = 1, 2, \cdots, n, j = 1, 2, 3, 4) \tag{18}$$

(4) The positive and negative ideal solutions are calculated as

$$\begin{aligned} V_{ij}^+ &= \max(z_{1j}, z_{2j}, z_{3j}, \ldots, z_{mj}) \\ V_{ij}^- &= \min(z_{1j}, z_{2j}, z_{3j}, \ldots, z_{mj}) \end{aligned} \tag{19}$$

(5) The Euclidean distances between each index and the positive and negative ideal solutions are calculated as

$$\begin{aligned} D_{ij}^+ &= \sqrt{\sum_{j=1}^{4} \left(V_{ij}^+ - V_{ij}\right)^2} \\ D_{ij}^- &= \sqrt{\sum_{j=1}^{4} \left(V_{ij}^- - V_{ij}\right)^2} \end{aligned} \tag{20}$$

(6) The comprehensive evaluation value is calculated as

$$C_{ij} = \frac{D_{ij}^-}{D_{ij}^+ + D_{ij}^-} \quad 0 < c_{ij} < 1 \tag{21}$$

(7) Optimization scheme selection: sorting is carried out according to the size of relative proximity, $C_{ij}$. When $C_{ij}$ is larger, the score of the evaluation object is higher and closer to the optimal value.

## 3. Results

### 3.1. Calibration and Validation of the AquaCrop-OSPy Model

The parameter calibration and validation of the model are used for the localization and application of the model. The parameter calibration of AquaCrop mainly takes *CC* and *BIO* as the reference standards. $R^2$, *NRMSE*, and *NSE* are used for the evaluation of the model performance. Figures 5 and 6, respectively, show the comparison between the measured and simulated values of *CC* and *BIO* in the test area from 2019 to 2021. The trial-and-error method is used to calibrate the relevant parameters. The final calibration results of parameters in AquaCrop-OSPy are shown in Table 5. The relevant parameters are essentially consistent with the values of Shan et al. (2022) for the AquaCrop constructed for corn in this experimental area [34].



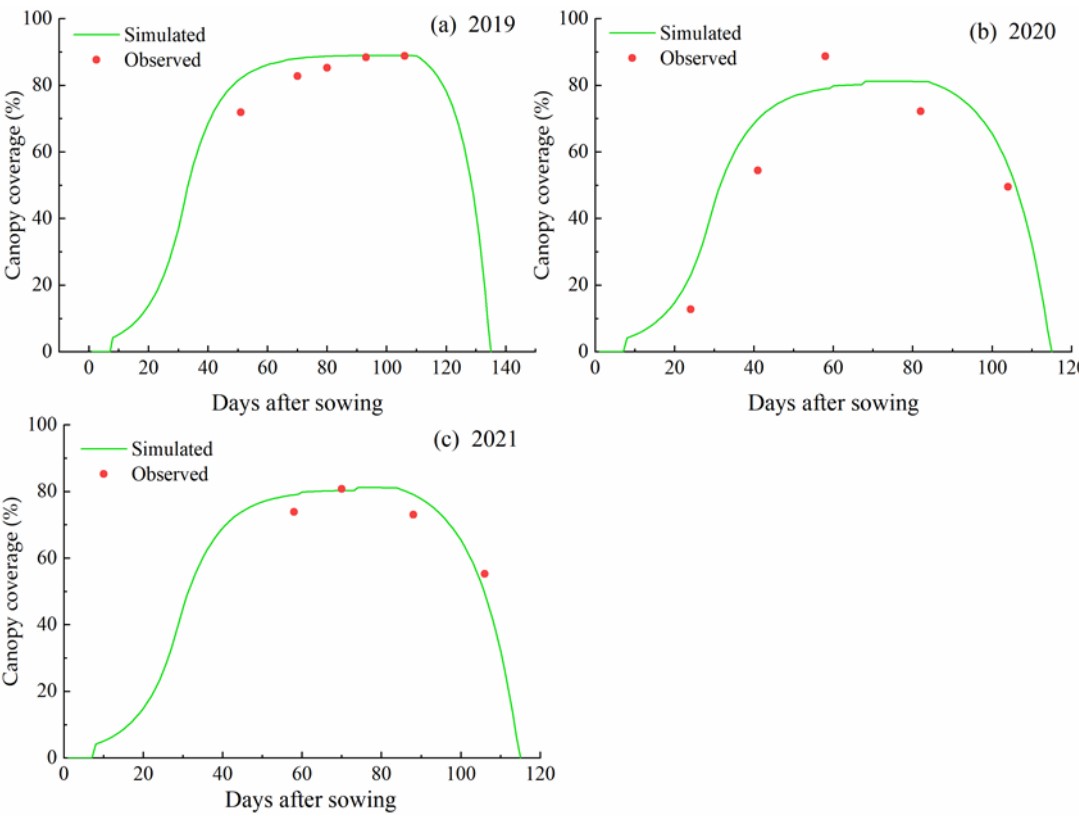

**Figure 5.** Modeled and observed Canopy coverage for maize during the 2019–2021 growing seasons.

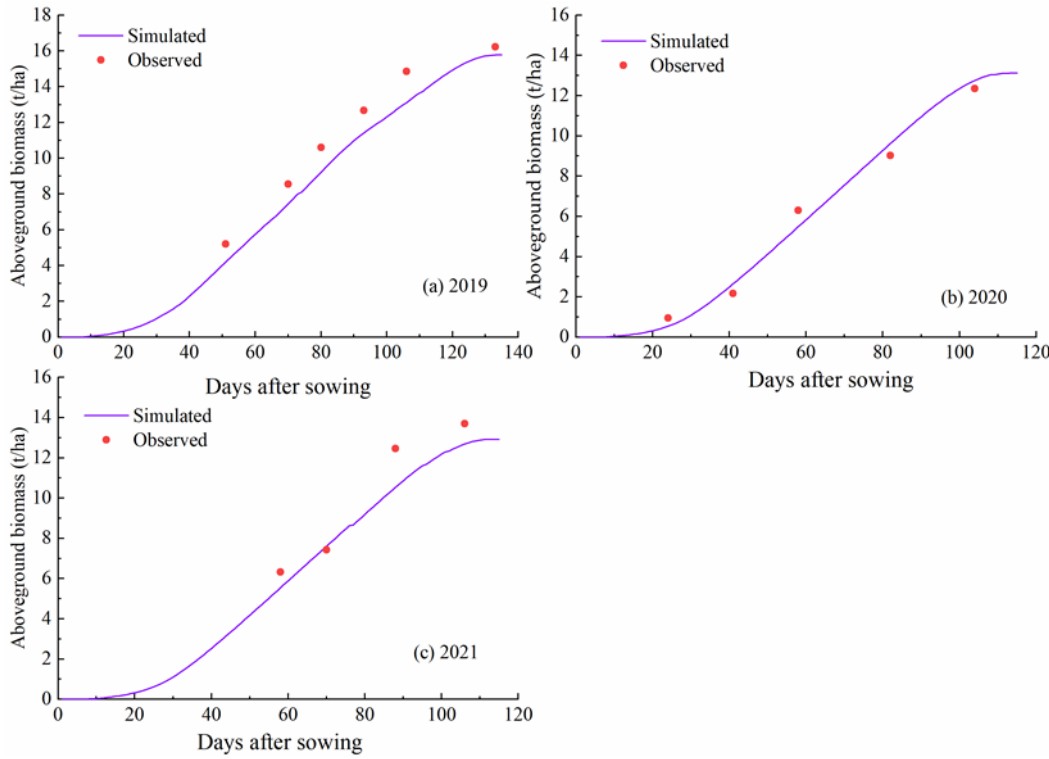

**Figure 6.** Modeled and observed Aboveground biomass for maize during the 2019–2021 growing seasons.

**Table 5.** Maize parameters in the ACOSP model.

| Parameter | Value | Parameter Description | Remarks |
|---|---|---|---|
| Name | Maize | | |
| PlantingDate | 6/15 | Planting date (mm/dd) | Measured |
| HarvestDate | 10/07 | Latest harvest date (mm/dd) | Measured |
| Emergence | 7 | Growing degree days from sowing to emergence | Measured |
| MaxRooting | 62 | Growing degree days from sowing to maximum rooting | Measured |
| Senescence | 84 | Growing degree days from sowing to senescence | Measured |
| Maturity | 115 | Growing degree days from sowing to maturity | Measured |
| HIstart | 63 | Growing degree days from sowing to start of yield formation | Measured |
| Flowering | 14 | Duration of flowering in growing degree days | Measured |
| Tbase | 8 | Base temperature below which growth does not progress (°C) | Recommended |
| Tupp | 30 | Upper temperature above which crop development no longer increases (°C) | Recommended |
| Zmin | 0.3 | Minimum effective rooting depth (m) | Measured |
| Zmax | 1.0 | Maximum rooting depth (m) | Measured |
| CCx | 0.85 | Maximum canopy cover | Measured |
| CDC | 0.094 | Canopy decline coefficient | Measured |
| CGC | 0.123 | Canopy growth coefficient | Calibrated |
| HI$_0$ | 0.35 | Reference harvest index | Calibrated |
| WP | 17 | Water productivity normalized for ET$_0$ and CO$_2$ (g/m$^2$) | Calibrated |
| p_up1 | 0.12 | Upper soil water depletion threshold for water stress effects on affect canopy expansion | Recommended |
| | 0.58 | Lower soil water depletion threshold for water stress effects on canopy expansion | Recommended |
| p_up2 | 0.14 | Upper soil water depletion threshold for water stress effects on canopy stomatal control | Recommended |
| p_up3 | 0.55 | Upper soil water depletion threshold for water stress effects on canopy senescence | Recommended |

In the "Remarks" column, "Calibrated" indicates that the values were calibrated using the data measured in 2019–2021, "Measured" indicates measured data, and "Recommended" indicates that the value was recommended in the AquaCrop manual.

The model is validated by calibrated parameters, and the validation results are shown in Table 6. We found that the model simulation overestimated *CC* and *BIO* under rainfall conditions. When the impact of water stress on crops is more serious, the simulation error of the model is greater. Li et al. (2014) obtained similar results for flax [35]. The model overestimates *CC* during crop development and does not underestimate *CC* during recession. Rainfall in the Yellow River Delta region mainly occurs in the late stage of maize growth, with fewer sunshine hours, which delays the aging of maize, and the model is sensitive to water stress in the recession period, in which the simulation speed of the recession rate is too fast. Similar findings were obtained in previous studies [36,37]. Although the simulation results are biased, the model can still effectively simulate the *CC* and *BIO* of maize under different irrigation treatments, thus establishing the relationship between *IW* and *Y*, *IWP*, and *WUE*.

**Table 6.** Evaluation of observed and simulated values for *CC* and *BIO*.

| Year | Particulars | Evaluation Parameters | | |
| --- | --- | --- | --- | --- |
| | | *NRMSE%* | *R²* | *NSE* |
| 2019 | *CC* | 8.90 | 0.87 | 0.74 |
| | *BIO* | 9.20 | 0.97 | 0.96 |
| 2020 | *CC* | 19.00 | 0.90 | 0.73 |
| | *BIO* | 10.2 | 0.96 | 0.98 |
| 2021 | *CC* | 7.0 | 0.92 | 0.73 |
| | *BIO* | 11.6 | 0.96 | 0.87 |

*CC*: canopy coverage; *BIO*: aboveground biomass.

*3.2. Optimization Results Using NSGA-III and TOPSIS-Entropy*

The Pareto frontier is the best specific method to solve conflicting multi-objective problems. Figure 7 shows the Pareto front and two-dimensional projection under the T2 and T6 scenarios in the Yellow River Delta region in the wet year, the normal year, and the dry year. The four objective functions are drawn in three-dimensional coordinates, where the *x*-axis in the three-dimensional coordinates represents *IW*, the *y*-axis represents *IWP*, and the *z*-axis represents *WUE*. The *Y* objective function is represented by different colors. *HV* and spacing are important indicators to evaluate the performance of the NSGA-III algorithm. The iterative process of *HV* and spacing is shown in Figures 8 and 9. All optimization problems tend to be stable in the 25th generation, representing the effective improvement in the scheme during the optimization process. The NSGA-III algorithm was used by Kelly et al. (2021) in their report, and scipy.optimize. fmin () was used to solve the nonlinear programming problem of the AquaCrop model with the linear programming function, which fell into the local optimal solution problem, so as to solve the problem of the irrigation strategy optimization of the model. Decision makers may have different decision bases and objectives rather than specifying specific weight combinations. Therefore, the entropy weight method is used to determine the target weight. Zhao et al. (2020) described the detailed procedure [38]. Then, the TOPSIS comprehensive evaluation method was used to rank the advantages and disadvantages of each evaluation objective according to different weight combinations. The weights of the four objectives in this study are shown in Table 7. Under the T2 and T6 irrigation scenarios, the target weight of min *IW* is the largest in the wet year, the normal year, and the dry year, and the target weight of max *IWP* is the smallest. The main reason is that the soil water content has been maintained at a high level, which increases the demand of crops for irrigation water, but the crop yield will not increase linearly with the increase in irrigation water. Under this influence, *IWP* will be seriously underestimated. In order to reduce this impact, the weight of *IWP* is assigned a low value. In rainy years, there is more rainfall, crop transpiration demand remains unchanged, and *IWP* and *WUE* are only affected by crop yield. At this time, their target weights are essentially equal. In the T6 irrigation scenario, the target weight of max *IWP* in the dry year is the largest, mainly due to less rainfall and higher irrigation frequency. In the scope of irrigation volume constraints, irrigation water has a greater impact on crop yield, so the weight of max *IWP* is higher.

The comprehensive TOPSIS-Entropy method is used to select the optimal solution from Pareto solutions with NSGA-III. The optimization methods for the first five T2 and T6 irrigation treatments in each typical year are shown in Table 8. The first scheme is called the best compromise scheme. For example, in the best compromise scheme under the T2 scenario in the wet year, the optimal irrigation volume of corn is 62.83 mm, the yield is 5.89 t/ha, the *IWP* value is 0.15, the *WUE* value is 1.90, and the optimal irrigation volume of corn in terms of the seedling emergence, canopy growth, maximum canopy, and senescence period is 73.49 mm, 93.76 mm, 89.13 mm, and 3.17 mm, respectively. It can be seen that the amount of irrigation in the senescence period of maize is less, mainly because the rainfall in the Yellow River Delta is concentrated in September, which is the time of corn maturity, and the amount of irrigation required for crops is less. According to the optimal compromise scheme obtained by TOPSIS, decision makers can formulate various irrigation schemes to meet their actual needs.

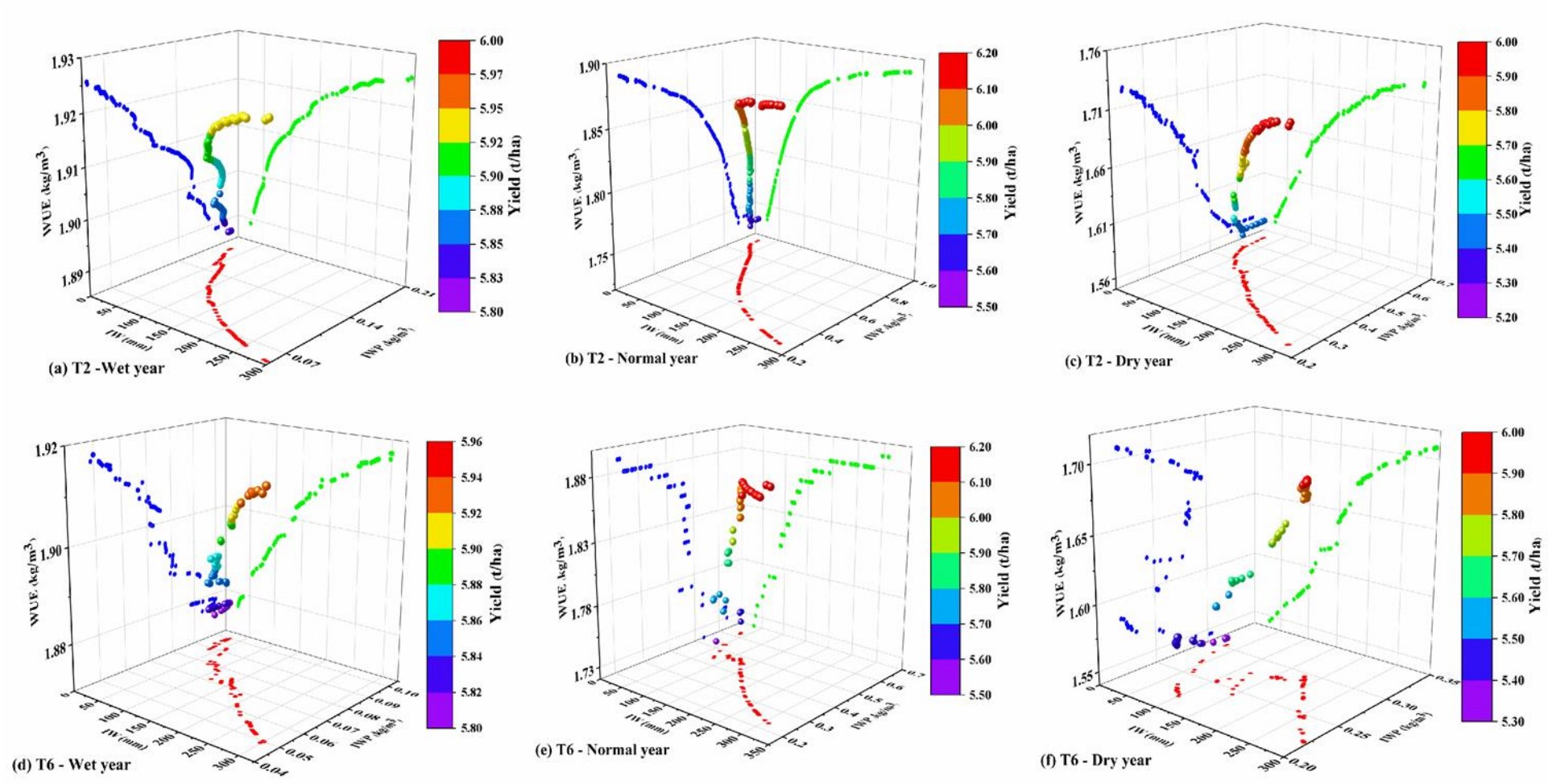

**Figure 7.** Scatter plots of optimization in the present study: Pareto front. *IW*, *IWP*, and *WUE* are the irrigation water, irrigation water production rate, and water use efficiency.

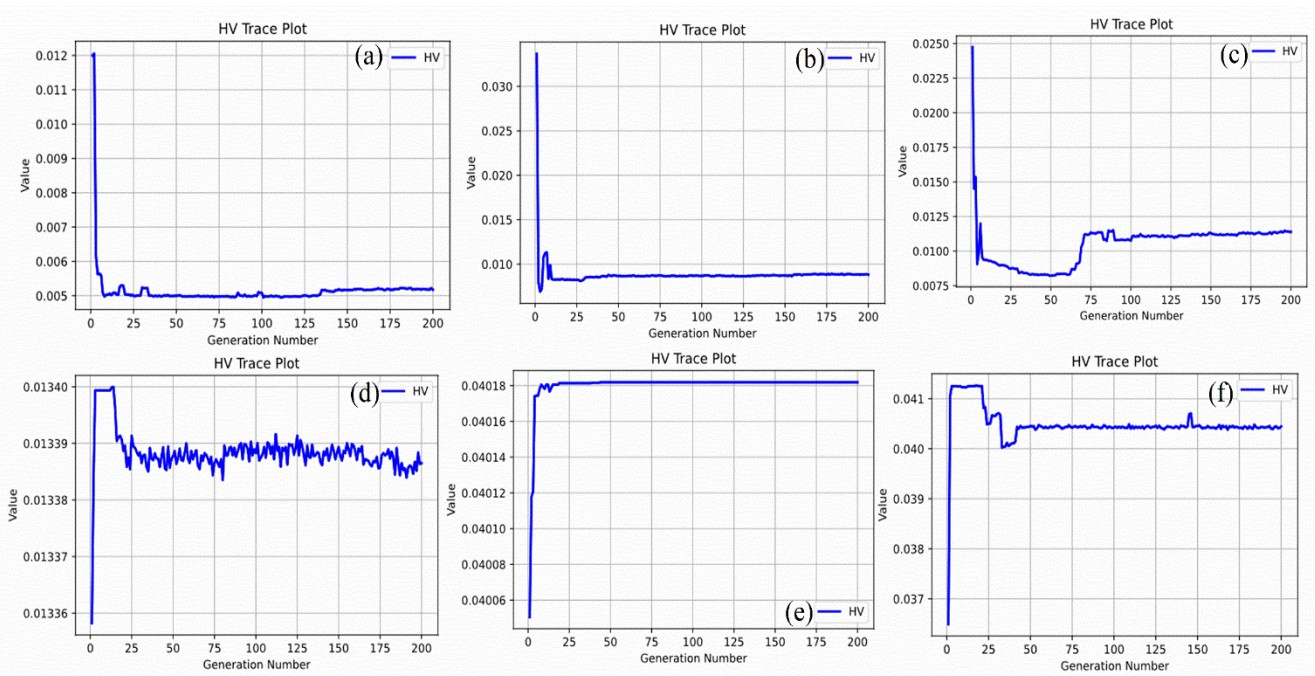

**Figure 8.** Performance of NSGA-III under T2 and T6. (**a**) *HV* curve for T2 in the wet year. (**b**) *HV* curve for T2 in the normal year. (**c**) *HV* curve for T2 in the dry year. (**d**) *HV* curve for T6 in the wet year. (**e**) *HV* curve for T6 in the normal year. (**f**) *HV* curve for T6 in the dry year.

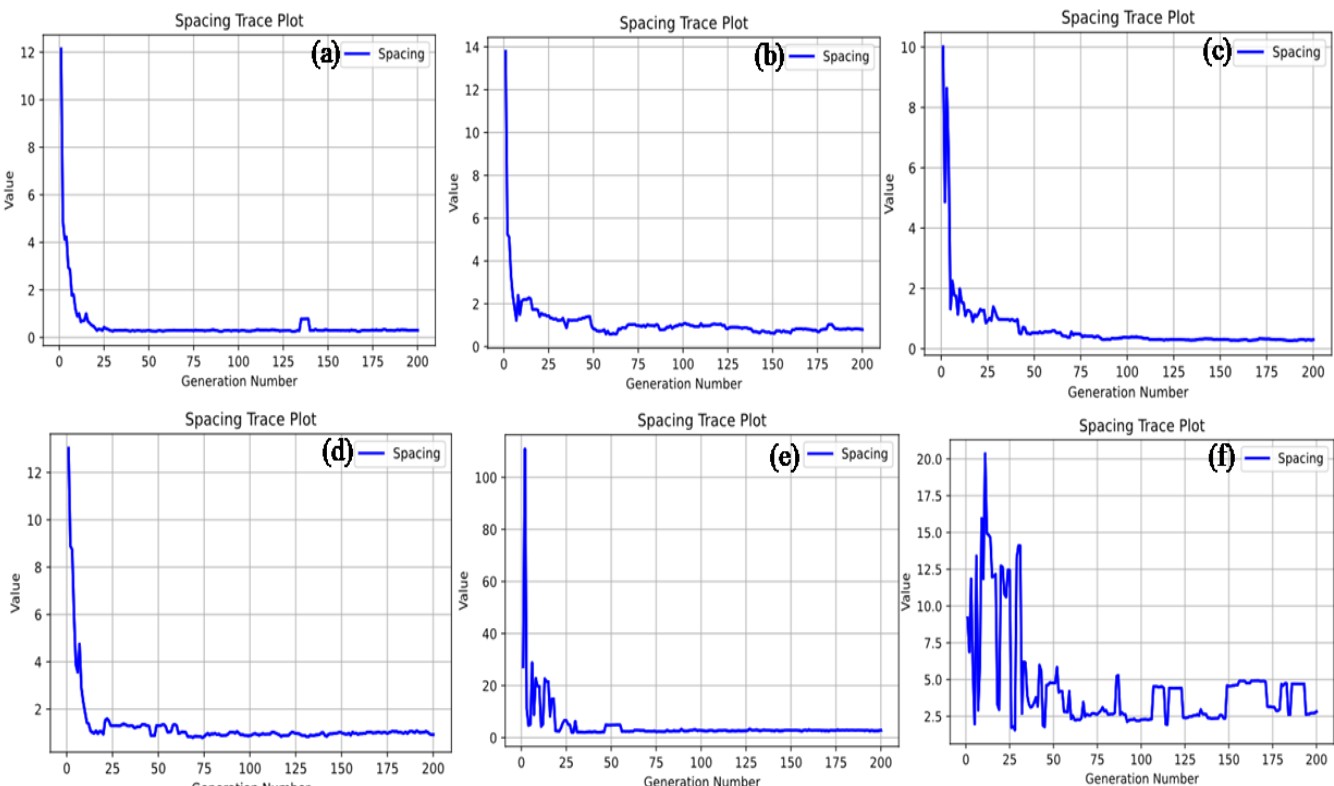

**Figure 9.** Performance of NSGA-III under T2 and T6. (**a**) Spacing curve for T2 in the wet year. (**b**) Spacing curve for T2 in the normal year. (**c**) Spacing curve for T2 in the dry year. (**d**) Spacing curve for T6 in the wet year. (**e**) Spacing curve for T6 in the normal year. (**f**) Spacing curve for T6 in the dry year.

**Table 7.** Weights in different scenarios using the entropy method.

| Scenario | Optimized Objective Ranking | Weight |
|---|---|---|
| T2—Wet year | f1 > f2 > f4 = f3 | [0.42, 0.20, 0.19, 0.19] |
| T2—Normal year | f1 > f2 > f4 > f3 | [0.48, 0.19, 0.15, 0.18] |
| T2—Dry year | f1 > f2 > f4 > f3 | [0.35, 0.25, 0.16, 0.24] |
| T6—Wet year | f1 > f2 = f4 > f3 | [0.42, 0.26, 0.06, 0.26] |
| T6—Normal year | f1 > f2 = f4 > f3 | [0.45, 0.25, 0.05, 0.25] |
| T6—Dry year | f3 > f1 > f2 > f4 | [0.19, 0.14, 0.56, 0.13] |

f1, f2, f3, and f4 represent minimum irrigation water (mm), maximum yield (t ha$^{-1}$), maximum irrigation water production rate (kg m$^{-3}$), and maximum water use efficiency (kg m$^{-3}$), respectively.

**Table 8.** Ranking of top five optimization solutions by TOPSIS.

| Scenario | Rank | Value | f1 | f2 | f3 | f4 | x0 | x1 | x2 | x3 | x4 |
|---|---|---|---|---|---|---|---|---|---|---|---|
| | 1 | 0.72 | 62.83 | 5.89 | 0.15 | 1.90 | 73.49 | 93.76 | 89.13 | 3.17 | 62.83 |
| | 2 | 0.72 | 36.33 | 5.85 | 0.17 | 1.89 | 49.34 | 93.03 | 52.98 | 29.24 | 36.33 |
| T2—Wet year | 3 | 0.69 | 171.52 | 5.93 | 0.08 | 1.92 | 84.74 | 93.55 | 55.67 | 36.38 | 171.52 |
| | 4 | 0.69 | 83.64 | 5.90 | 0.13 | 1.90 | 69.25 | 93.56 | 0.000 | 41.26 | 83.64 |
| | 5 | 0.69 | 244.17 | 5.95 | 0.06 | 1.92 | 89.11 | 94.53 | 86.01 | 47.21 | 244.17 |
| | 1 | 0.78 | 72.60 | 5.96 | 0.72 | 1.83 | 20.06 | 84.95 | 96.31 | 94.45 | 72.60 |
| | 2 | 0.78 | 85.62 | 6.00 | 0.67 | 1.85 | 39.73 | 85.58 | 87.58 | 99.90 | 85.62 |
| T2—Normal year | 3 | 0.78 | 68.76 | 5.94 | 0.74 | 1.83 | 41.79 | 84.82 | 81.16 | 74.33 | 68.76 |
| | 4 | 0.78 | 121.83 | 6.07 | 0.53 | 1.87 | 31.80 | 86.99 | 94.38 | 78.31 | 121.83 |
| | 5 | 0.78 | 55.32 | 5.87 | 0.80 | 1.80 | 27.10 | 84.88 | 87.87 | 15.47 | 55.32 |
| | 1 | 0.76 | 31.08 | 5.45 | 0.56 | 1.57 | 25.40 | 53.57 | 97.83 | 88.12 | 31.08 |
| | 2 | 0.75 | 125.83 | 5.80 | 0.41 | 1.67 | 74.21 | 83.37 | 86.30 | 89.70 | 125.83 |
| T2—Dry year | 3 | 0.74 | 90.64 | 5.69 | 0.46 | 1.64 | 73.47 | 78.71 | 85.25 | 96.69 | 90.64 |
| | 4 | 0.74 | 247.17 | 5.97 | 0.28 | 1.72 | 83.84 | 89.63 | 93.49 | 96.89 | 247.17 |
| | 5 | 0.74 | 171.55 | 5.87 | 0.35 | 1.69 | 74.17 | 86.85 | 79.30 | 99.16 | 171.55 |
| | 1 | 0.93 | 132.23 | 5.89 | 0.07 | 1.90 | 0 | 16.53 | 0.15 | | |
| | 2 | 0.93 | 141.88 | 5.90 | 0.07 | 1.90 | 0 | 12.90 | 0.13 | | |
| T6—Wet year | 3 | 0.93 | 273.24 | 5.93 | 0.05 | 1.92 | 0 | 11.88 | 0.08 | | |
| | 4 | 0.93 | 184.26 | 5.92 | 0.07 | 1.91 | 0 | 16.75 | 0.11 | | |
| | 5 | 0.93 | 278.71 | 5.94 | 0.05 | 1.92 | 0 | 12.12 | 0.08 | | |
| | 1 | 0.96 | 269.00 | 6.13 | 0.26 | 1.88 | 0 | 7.70 | 0.09 | | |
| | 2 | 0.96 | 246.00 | 6.12 | 0.28 | 1.88 | 0 | 8.47 | 0.10 | | |
| T6—Normal year | 3 | 0.96 | 266.00 | 6.13 | 0.26 | 1.88 | 0 | 7.61 | 0.09 | | |
| | 4 | 0.96 | 118.00 | 6.00 | 0.48 | 1.84 | 0 | 13.12 | 0.20 | | |
| | 5 | 0.96 | 278.00 | 6.13 | 0.25 | 1.88 | 0 | 7.73 | 0.08 | | |
| | 1 | 0.67 | 148.32 | 5.70 | 0.28 | 1.64 | 0 | 18.54 | 0.21 | | |
| | 2 | 0.60 | 252.63 | 5.91 | 0.25 | 1.70 | 2.39 | 20 | 0.16 | | |
| T6—Dry year | 3 | 0.60 | 255.91 | 5.91 | 0.25 | 1.70 | 2.60 | 19.87 | 0.16 | | |
| | 4 | 0.60 | 97.11 | 5.50 | 0.22 | 1.58 | 4.41 | 0 | 0.14 | | |
| | 5 | 0.60 | 86.29 | 5.48 | 0.23 | 1.58 | 3.92 | 0 | 0.14 | | |

f1, f2, f3, and f4 represent minimum irrigation water (mm), maximum yield (t ha$^{-1}$), maximum irrigation water production rate (kg m$^{-3}$), and maximum water use efficiency (kg m$^{-3}$), respectively. x0, x1, x2, x3, and x4 in the T2 treatment represent crop emergence, canopy growth, maximum canopy, senescence, and irrigation amount in the growth period, respectively. x0, x1, and x2 in the T6 treatment represent the amount of I$_{rain}$, I$_{no-rain}$, and *TAW* (%) in the next 10 days, respectively.

### 3.3. Responses to Irrigation Strategy Optimization under Different Scenarios

The *IW*, *Y*, *IWP*, and *WUE* values of maize under six irrigation scenarios (the best compromise scheme for T2 and T6 is selected) are shown in Figure 10. It can be seen that the net irrigation can obtain a yield of 5.87 t/ha yield without irrigation in the wet year, only 0.02 t/ha lower than the maximum yield (T2), and the corresponding irrigation amount is reduced by 62.83 mm. The optimal *TAW* irrigation water demand in the normal and dry year is the least, but outputs of 5.96 t/ha and 5.45 t/ha output are obtained, which are 0.17 t/ha and 0.3 t/ha lower than the highest output (T6 and T5), respectively. The irrigation amount of 180 mm is the largest in each typical hydrological year, but the yield is the lowest.

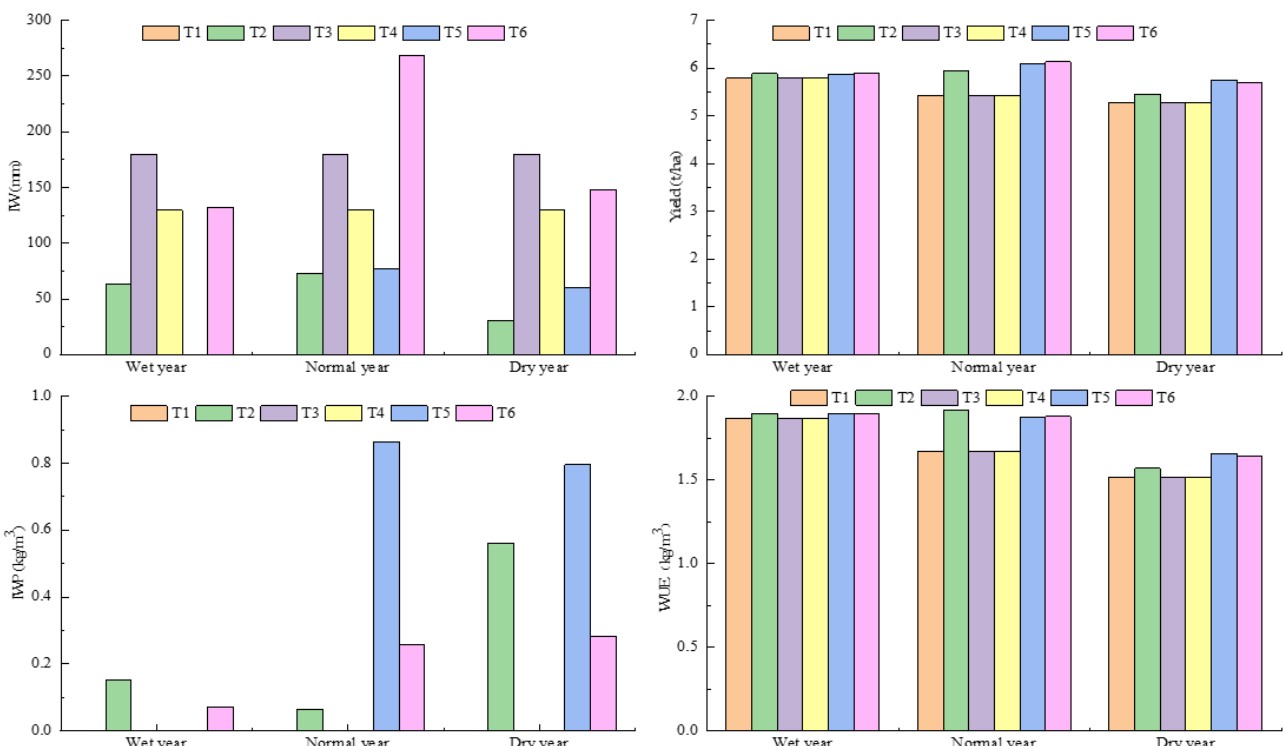

**Figure 10.** Maize *IW* (mm), *Y* (t ha$^{-1}$), *IWP* (kg m$^{-3}$), and *WUE* (kg m$^{-3}$) under different irrigation scenarios. Note: T1, rainfed conditions; T2, optimal *TAW* (%) irrigation; T3, irrigation in seedling, jointing, and grouting stages, with a total amount of 180.0 mm; T4, irrigation in seedling, jointing, and grouting stages, with a total amount of 130.0 mm; T5, net irrigation; and T6, optimal irrigation considering the weather conditions. *IW*, *IWP*, and *WUE* are the irrigation water, irrigation water production rate, and water use efficiency.

Comparing the five irrigation scenarios, it can be seen that the T2 irrigation scenario has significant advantages in improving *Y*, *IWP*, and *WUE*. In the wet year, *Y*, *IWP*, and *WUE* are the best of the five irrigation scenarios under the T2 irrigation scenario. In the normal and dry year, the *Y* and *WUE* of T2 irrigation are higher, which is closely related to the T2 irrigation method. T2 irrigation is triggered by monitoring the soil water content at each growth stage of crops, which is lower than the threshold of the growth stage, consistent with the current precision irrigation scheme used in smart agriculture.

In addition, *IWP* can comprehensively reflect the agricultural production level, irrigation engineering status, and irrigation management level of the irrigation area. Our analysis of the *IWP* of six irrigation scenarios in three typical hydrological years shows that net irrigation is significantly higher than other irrigation treatments in the normal and dry years, while the *IWP* of net irrigation in the wet year is 0, which may be because the rainfall in the wet year of the region met the requirement for the normal growth of corn. The effect of irrigation on crop yield is not obvious. *WUE* analysis of three typical hydrological years found that the water use efficiency of net irrigation was at a higher level compared with other irrigation scenarios. According to the comprehensive analysis of *IW*, *Y*, *IWP*, and *WUE*, the impact of irrigation in high-water years on crops in the Yellow River Delta is relatively small, and net irrigation in normal- and low-water years is the best irrigation scheme.

### 3.4. Irrigation Demands under Different Climate Change Scenarios

RCPs are a series of comprehensive concentration and emission scenarios, which are used as the input parameters of climate change prediction models under the influence of human activities in the 21st century [39]. On the basis of keeping the parameters of the AquaCrop soil, crop, field management, and groundwater modules unchanged, the

climate change prediction changes in accordance with the meteorological data module to predict future corn yield changes. In 2014, the Fifth Coupled Model Intercomparison Project Phase 5 (CMIP5) provided the latest data for predicting future climate change. The researchers proposed four typical greenhouse gas concentration paths (RCPs) as scenarios for predicting future climate, namely RCP2.6, RCP4.5, RCP6.0 and RCP8.5 [40,41]. The RCP2.6 path simulates the scenario that the global temperature will rise by less than 2 degrees Celsius by 2100 compared with that before the industrialization era. The RCP8.5 path simulates a temperature rise of 5 degrees Celsius by 2100. Research shows that because RCP4.5 has a higher priority than RCP6.0, and RCP2.6 is the most ideal of the four emission scenarios, the research on RCP6.0 is of little significance [42–45]. Therefore, only the estimated results under the climate models RCP4.5 and RCP8.5 are selected.

AquaCrop-OSPy includes a built-in functionality for externally generating future climate scenarios using LARS-WG, which is a widely used stochastic weather generator for agroclimatic impact assessments. Meteorological data are transformed using the "prepare_lars_weather" function, which includes functions for calculating the evapotranspiration required for AquaCrop [15]. Projections generated for the RCP4.5 and RCP8.5 emission scenarios in 2021–2040, 2041–2060, and 2061–2080 based on the LARS-WG climate model output are shown in Figure 11. Therefore, the climate prediction results are combined with the verified AquaCrop-OSPy model to simulate the corn yield change under future climate change scenarios [15,46,47], and the reasons for the crop yield change are analyzed in combination with the climate change trend in the growth period, providing a certain theoretical basis for local agricultural production management. Assuming that a constant irrigation water threshold of 70% will be maintained throughout the year in the future, we analyzed the crop yield and irrigation water demand. The results show that in future climate scenarios, higher $CO_2$ concentrations will increase the yield and slightly reduce the demand for irrigation water. The prediction of increased crop water productivity is consistent with the findings obtained by Kelly and Foster (2021) [15]. Compared with the 1986–2015 baseline, yields will be lower under future climate change scenarios and irrigation water consumption will be slightly higher. Under future climate change scenarios, global warming will reduce rainfall in the study region, thereby affecting crop yields and irrigation water usage.

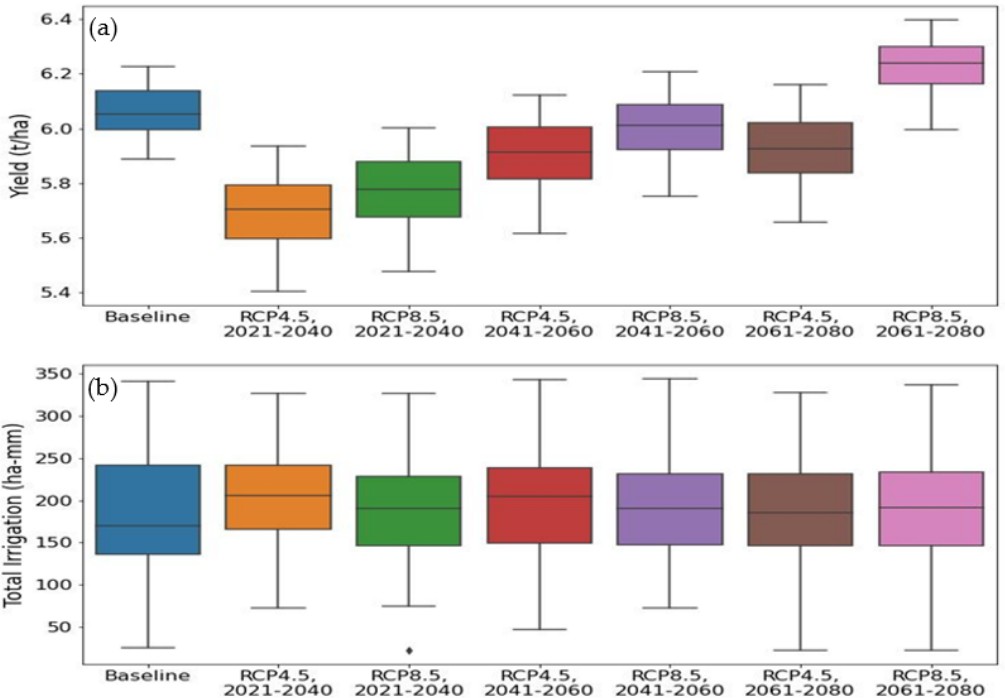

**Figure 11.** Comparison of (**a**) yield and (**b**) total irrigation of maize crops in Dongying City, Shandong Province, China, under baseline climate conditions (1986–2015) and various future climate scenarios.

## 4. Discussion

The simulation optimization model coupled with AquaCrop and the NSGA-III algorithm established using the Python language is used to simulate the *IW*, *Y*, *IWP*, and *WUE* of corn under three typical hydrological years and six irrigation strategies in the Yellow River Delta region, and the model parameters are calibrated and verified through three years of field test data. The results show that after calibration, the model parameters could accurately simulate the *CC* ($R^2 \geq 0.87$, $NRMSE \leq 19\%$) and *BIO* ($R^2 \geq 0.96$, $NRMSE \leq 11.6\%$) of corn, and the simulation effect under the irrigation scenario was better than that under rain-fed conditions, which may be related to the uneven distribution of rainfall in the region leading to water stress in rain-fed corn. The simulation accuracy of AquaCrop under water stress decreased, which was confirmed by Sandhu and Irmak et al. [3,34].

In the T2 and T6 irrigation strategies, the optimal solution is selected from the Pareto solution of NSGA-III by TOPSIS-Entropy. The optimal compromise yield of the T2 irrigation scheme in the wet year, the normal year, and the dry year is 5.89 t/ha, 5.96 t/ha, and 5.45 t/ha, respectively, and the irrigation volume is 62.83 mm, 72.60 mm, and 31.08 mm, respectively. The best compromise yield of the T6 irrigation scheme in the high-water year, normal-water year, and low-water year is 5.89 t/ha, 6.13 t/ha, and 5.70 t/ha, respectively, and the irrigation volume is 132.23 mm, 269.00 mm, and 148.32 mm, respectively. It can be seen that the T2 and T6 treatments are not the best in the *IW* or *Y* rankings after TOPSIS-Entropy is selected, but decision makers (such as farmers and policy specifiers) can choose the specific optimal scheme for a certain target. NSGA-III uses the AquaCrop model suite as the optimization objective function to find a novel solution for irrigation optimization in irrigation areas [20,22,46].

It can be seen from the performance of the four objectives under different irrigation scenarios that *IW* has a significant impact on maize *Y*, *IWP*, and *WUE*. Maize *Y* will not increase linearly with the increase in *IW*. Excessive irrigation will not only waste water resources, but also lead to a reduction in crop yield, which is consistent with the research results of Markovic et al. [47]. In addition, we compared T2 with T3 and T4 and found that in the same typical hydrological year, *IWP* and *WUE* were lower with the increase in irrigation volume, because the increase in irrigation volume was greater than the increase in yield. According to the analysis of yield and *WUE* in three typical hydrological years, the yield and *WUE* in dry years are the lowest, which may be due to the excessive water stress, limiting crop growth and water use [48].

Assuming that the constant irrigation water threshold of 70% will be maintained throughout the year in the future, under the RCP4.5 and RCP8.5 scenarios, the corn yield will show a significant downward trend, and the total irrigation water volume will increase significantly, which may be due to the reduction in the crop growth period and dry matter accumulation time caused by a rise in temperature [49]. This research shows that reasonable water and fertilizer management and crop variety improvement can also offset the negative effects of climate change on yield, thus increasing the amount of irrigation during crop growth [50,51].

## 5. Conclusions

Based on three years of field test data and 55 years of meteorological data in the Yellow River Delta, the growth performance of maize under six irrigation scenarios in three typical hydrological years was simulated using the coupled model of AquaCrop-OSPy and NSGA-III. The results show that the model can accurately simulate the canopy coverage and aboveground biomass of maize under irrigation conditions. Comparing the four objectives of six irrigation strategies under three typical hydrological years, it is found that net irrigation is the best irrigation method in the test area in the normal and dry year, and that the rainfall in the wet year can meet the water demands of maize during the growth period. The coupling of AquaCrop-OSPy and NSGA-III alleviates the difficulties of large-scale farmland management [11,20], but does not directly solve issues such as those relating to nutrient application, environmental impact, and economic cost.

Future research should be carried out to realize the sustainable development of large-scale farmland. By using LARS-WG to generate future climate scenarios from the outside and the AquaCrop-OSPy model to simulate and analyze the impact of future climate change on maize irrigation and yield in the experimental area, it was found that the increase in annual average temperature in the future will have a negative impact on maize yield and irrigation, and the prediction of maize irrigation and yield in 2021–2080 reflects the change trend of future maize irrigation and yield, to a certain extent. However, the simulation results are based on a constant irrigation water threshold of 70% in future growth periods, without considering the impact of water stress, diseases, and insect pests on corn yield. There are many uncertainties in future climate change scenarios, and the uncertainty of crop models is also a key factor affecting crop yield, which needs further study.

**Author Contributions:** Methodology, W.M.; formal analysis, L.S.; investigation, G.L.; resources, Y.S. (Yuyang Shan); Software, S.T.; data curation, S.T.; writing—original draft preparation, G.L.; writing—review and editing, Y.S. (Yuyang Shan); visualization, L.S.; supervision, L.S., Q.W. and Y.S. (Yan Sun); project administration, Y.S. (Yan Sun). All authors have read and agreed to the published version of the manuscript.

**Funding:** This research was funded by the National Natural Science Foundation of China (51979220, 52009039, 52209056), the Major Science and Technology Projects of the XPCC (2021AA003-2), the Natural Science Foundation of Xinjiang Uygur Autonomous Region (2022D01B70), and the Shaanxi Provincial Department of Education Special Scientific Research Project (21JK0783).

**Institutional Review Board Statement:** Not applicable.

**Informed Consent Statement:** Not applicable.

**Data Availability Statement:** The data presented in this study are available on request from the corresponding author.

**Conflicts of Interest:** The authors declare no conflict of interest. The funders had no role in the design of the study; in the collection, analyses, or interpretation of data; in the writing of the manuscript; or in the decision to publish the results.

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
