# Peer review of "Optimizing the Maize Irrigation Strategy and Yield Prediction under Future Climate Scenarios in the Yellow River Delta"

_agronomy, doi:10.3390/agronomy13040960_

Round 1

Reviewer 1 Report

To improve the quality of the paper, it is necessary to consider following requirements:

 -   Page 1 the title “Introduction” must be transferred to the next page.

-          - The numbering of pages must be verified :

o   There is many pages presented the same number.

o   Some pages didn’t present a number

-          - It necessary to check the figures and tables numbers.

-          - Page 3 : the page presents figures and tables not cited in the text.

-          - Page 5: it is necessary the revise the part “2.5…” line 216.

-          - Page 5 table 3: it is necessary to present e commentary of the table.

-          - Page 5, line 245: the title “Maximum Yeild” must be transferred to the next page.

-          - Page 12: the table 6 not cited in the text.

-          - Page of figure 8 : the figure 8 not cited in the text.

-          - Page noted 2 of 24: the title of figure 10 must be transferred to the page before.

Author Response

We gratefully thank you for taking the time to make constructive remaks and useful suggestions, which has significantly improved the quality of the manuscript and enable us to improve the manuscript. Each proposed revision and comment you put forward has been accurately taken into account. Please refer to the attachment for point by point response and revision of your comments and suggestions.

Reviewer 2 Report

 The manuscript submitted to Agronomy addresses a topic of interest to the journal's readers. However, it requires significant revisions. The authors should focus on improving the manuscript's presentation, shortening it, and enhancing its writing style.

To ensure clarity, the authors should define all abbreviations when they are first used.

In the Materials and Methods (MM) section, authors should describe how canopy cover was determined throughout the cycle, including the processing software used.

Lines 416-426 and Table 6 should also be included in the MM section.

Lines 428-453 should focus on the results and discussion, while all other information should be placed in the MM section. T

Authors must take great care to communicate the results and conclusions of their work effectively.

Overall, the manuscript requires significant improvements to be suitable for publication in Agronomy.

Author Response

(The authors gave the same response as above.)

Reviewer 3 Report

This work is very interesting and ready to be published after a minor modification. Authors have to work hard and finish important tasks. There are some notes, like

1- The abstract section is very large, please summarize it.

2- Table 2, it needs to make some adjustments to the format.

3- Figure 2, please arrange the sub-figures of (a), (b), and (c) according to the year; 1987, 1995, and 2011

4- In line 154, “the Pearson type III distribution method is described as follows”. Rewrite then put :

5- Again, Figure 5, please arrange the sub-figures according to the year; 1961, 1982, and 1993

6- Based on MDPI guidelines, revise the concluding section.

7- There are many problems with the grammar of English. Please read through the text again.

Author Response

(The authors gave the same response as above.)

Reviewer 4 Report

Hi Authors,

Thanks for this work. You have to follow the journal's guidelines for how you can write the reference inside the paper. please rewrite all the references. 

Table 2. you have to redo it. 

Table 7&8. you have to redo all the numbers such as 0.7222 to 0.72. 

Best wishes

Author Response

(The authors gave the same response as above.)

Reviewer 5 Report

Comments and Suggestions for Authors

The paper titled ‘Optimizing the maize irrigation strategy and yield prediction under future climate scenarios in the Yellow River Delta’ aims to give a valuable tool for stakeholders and decision-makers regarding irrigation strategies in light of climate change.

The abstract is informative, however, I suggest shortening it.

What I missed in the Introduction section is the information about the frequency of so-called by Authors' typical years in the multi-annual period. How often do the dry years appear in this region, what is the frequency of the other types of years?

The section on Materials and Methods is, in my opinion, artificially expanded. Due to it the paper is too long and misses clarity. Some of the indicators described in detail by the Authors are commonly used and there is no need to call them in detail, appropriate references would be good enough.

The results section needs to be improved. In some sections, the Authors just duplicate the information from tables, while missing in-depth analysis of the results.

The conclusion section seems to be a duplication of results, while it is supposed to be an exaggeration of the main findings.

The citations of sources do not go along the references list. There are some positions, which are not cited in the paper. Also, the way the references being cited in the manuscript do not follow the Journal rules.

I recommend the Authors make some efforts to clarify the paper and tidy up the references. Only under these conditions, I would recommend publishing it in the Journal.

I also found some minor incorrections, which need to be changed/corrected:

L46: The authors write about maize and therefore why do they characterize wheat in the introduction section? No information, what is the share of maize in the crops in the region?

L 132: unnecessary left parenthesis : (I1

L134: Jointing – a lower letter will be fine as for another stages names

L389: in The Table 6 heading the Authors mention:  yield (Y), while there is no data on yield in this table

L 459: in the heading of Figure 10 there are no explanations for graphs d, e and f

L480: in the heading of Table 7 there is no explanation of what f1, f2, f3, and f4 means

L481: in the heading of Table 8 there is no explanation of what x1,x2 x3, x4 means

L503: in the header of Figure 11, there are explanations of the Y-axis descriptions of three of the 4 figures presented. Why is there no description for the left/top IW drawing?

L542: The increase in yields is not only affected by CO2 and the change in its concentration in the atmospheric air. Plant productivity is affected by several factors, some of which we can control, such as breeding progress and applied fertilization. Therefore, it does not seem justified to explain changes in crop yield solely by modeling changes in CO2 concentration in the future.

Author Response

(The authors gave the same response as above.)

Round 2

Reviewer 1 Report

All required recommandations were done.  

Author Response

Thanks to your suggestions, the revised article is better. We appreciate your help.

Reviewer 5 Report

Thanks to the Authors for responding to my comments. However, please note two points:

 ad. Response 2

Please re-check the frequency of wet, normal and dry years as the sum of the numbers cited by the authors is 150% in total, while it should be 100%.

ad. Response  6

Please check how to cite the references in text of the manuscript submitted to the journal Agronomy.

Author Response

Thank you for your comments. I have explained and revised your question. Please refer to the attachment for point by point response and revision of your comments and suggestions.
